# Cancer Screening: Present Recommendations, the Development of Multi-Cancer Early Development Tests, and the Prospect of Universal Cancer Screening

**DOI:** 10.3390/cancers16061191

**Published:** 2024-03-18

**Authors:** Laurenția Nicoleta Galeș, Mihai-Andrei Păun, Rodica Maricela Anghel, Oana Gabriela Trifănescu

**Affiliations:** 1Department of Oncology, Carol Davila University of Medicine and Pharmacy, 020021 Bucharest, Romania; laurentia.gales@umfcd.ro (L.N.G.); rodica.anghel@umfcd.ro (R.M.A.); oana.trifanescu@umfcd.ro (O.G.T.); 2Department of Medical Oncology II, Prof. Dr. Al. Trestioreanu Institute of Oncology, 022328 Bucharest, Romania; 3Department of Radiotherapy II, Prof. Dr. Al. Trestioreanu Institute of Oncology, 022328 Bucharest, Romania

**Keywords:** universal cancer screening, multi-cancer early detection, biomarker, cancer screening

## Abstract

**Simple Summary:**

This article is intended to serve as a roadmap for scientists, steering them toward fruitful avenues of investigation. Moreover, the review makes a compelling argument for the implementation of universal cancer screening and multi-cancer early detection screening. It synthesizes evidence supporting the efficacy of such screenings, emphasizing their potential to detect various cancers at early, more treatable stages. By presenting a well-reasoned case, the review contributes to the discourse on the importance of widespread screening, urging healthcare providers and policymakers to consider its adoption for improved cancer detection and outcomes.

**Abstract:**

Cancer continues to pose a considerable challenge to global health. In the search for innovative strategies to combat this complex enemy, the concept of universal cancer screening has emerged as a promising avenue for early detection and prevention. In contrast to targeted approaches that focus on specific populations or high-risk individuals, universal screening seeks to cast a wide net to detect incipient malignancies in different demographic groups. This paradigm shift in cancer care underscores the importance of comprehensive screening programs that go beyond conventional boundaries. As our understanding of the complex molecular and genetic basis of cancer deepens, the need to develop comprehensive screening methods becomes increasingly apparent. In this article, we look at the rationale and potential benefits of universal cancer screening.

## 1. Introduction

In 2020, the World Health Organization (WHO) statistics showed that cancer exacts an alarming toll, one in six deaths per year worldwide, with an alarming increase in overall mortality globally [1,2,3].

Cancer mortality can be prevented through cancer screening programs, which have a considerable impact on reducing cancer-specific mortality by selecting pre-symptomatic cases and driving more extensive diagnostic procedures and finally prompt treatment. For instance, breast cancer mortality in the United States declined by 53.2% between 1975 and 2014, with similar trends observed in Europe, which is attributed to both screening and improved treatments [4,5,6]. There is a need to avoid late detection, which is when cancer is diagnosed when it has already progressed beyond any potential curative options, or metastasized, where it is most often fatal, which in an estimated 15% of cases could have been avoided by early disease detection.

Among the countries with the highest per capita healthcare spending, cancer screening recommendations are relatively consistent, including breast, cervical, colorectal, prostate, skin, and lung cancer screening. However, differences in healthcare utilization exist between countries (and even within them), with a particular impact on cancer screening recommendations [4,7]. As such, there is a lack of universal consensus on which cancers should be screened for in the general population, with important malignancies left unchecked. For instance, cancers of the lung, pancreas, esophagus, stomach, and ovary are most often diagnosed in the metastatic setting, with a lack of population-based methods for screening, which is the case for up to 60% of cancer-related deaths where nationwide strategies for screening do not exist [8,9,10].

Current screening methods have increased survival rates, but they also have some drawbacks, including the risk of overdiagnosis, false-positive results, exposure to radiation, invasiveness leading to a decrease in patience compliance, and lack of support from the general public to undergo these investigations [10,11,12,13,14,15,16].

Furthermore, the current perspective on cancer screening might be conceptually inexact, as detection relies on “one organ at a time” intervention (Figure 1). Thus, most screening strategies are ineffective due to the high cost of a method designed only for one malignancy, thus excluding the potential diagnosis of other cancers that the patient might be suffering from. In contrast, universal cancer screening is a concept that predicts a future where detection is directed at multiple organs and multiple tumor types simultaneously with a single, potentially non-invasive method. So far, myriad multi-cancer early detection (MCED) methods have been proposed, gathering interest from clinicians to improve cancer detection and outcomes [7,17,18].

In this article, our goal is to provide insights on new developments in MCED biomarker research by sharing the latest results and directions that bring us closer to universal cancer screening, a concept that envisions testing persons from the general population to detect early cancers, no matter which type or site, in their pre-symptomatic phase, with a single, cost-effective, minimally invasive test.

## 2. Status Quo of Cancer Screening

Before we present the updates in screening biomarker research, we want to briefly present the current materials and methods for cancer screening. It is important to note that there is still a long way to go until consensus is reached over the targeted population (risk categories, age, frequency of testing) and that these tests have different categories of recommendation (Table 1).

### 2.1. Colorectal Cancer

Colorectal cancer benefits from some of the best screening tools available, including stool tests and colonoscopies.

Stool-based tests include the guaiac fecal occult bleeding test (g-FOBT), fecal immunochemical test (FIT), and stool DNA test (DNA-FIT). The g-FOBT and FIT tests detect blood in stool, relying on the premise that cancers will cause ruptures of small vessels, leading to clinically insignificant bleeding [20,21]. Although these tests have high sensitivity and specificity, it is worth noting that they have lower sensitivity for pre-malignant lesions such as adenomas (especially smaller ones), which do not invade the blood vessels in the submucosa [22,23].

Detection (and also resection) of precancerous lesions is only possible through colonoscopy, which is considered the gold standard for colorectal cancer screening due to its high sensitivity and specificity, relative safety, and demonstrating reduction of overall colorectal cancer mortality [24]. Possible limitations to this test include a certain degree of invasiveness and dependence on the training level and experience of the gastroenterologist, leading to discrepancies in the outcomes shown by different studies [25].

There are no validated serum biomarkers used in colorectal cancer screening. Serum CEA and CA19-9 markers are not used in screening due to low sensitivity and specificity, but are validated for prognosis and post-resection follow-up [16,26].

The United States Preventive Services Taskforce (USPSTF) highly recommends colorectal cancer screening for patients aged 50–75 (grade A). For other age groups, the USPSTF suggests screening for patients aged 45–49 (grade B) and also 76–85 (grade C—based on life expectancy and patient risk). Screening modalities include colonoscopy every 10 years, g-FOBT or FIT yearly, with stool DNA-FIT every 1–3 years, and flexible sigmoidoscopy every 5 years (or flexible sigmoidoscopy every 10 years plus FIT every year) [27]. The American Cancer Society (ACS) also recommends screening for adults aged 45–75, selective screening for those 76–85, and no screening for those over 85. For patients under the age of 45, high-risk features may cause earlier intervention. Patients with relatives diagnosed with colorectal cancer should be screened 10 years before the diagnosis age of the relative. Patients with abdominal or pelvic irradiation history should begin screening 5 years post-intervention or starting at age 30. Patients suffering from inflammatory bowel disease also require colonoscopies at most 8 years after diagnosis, repeated every 1–3 years. Most importantly, patients with removed polyps during colonoscopy should repeat the test after 3 years maximum, possibly earlier if malignant or high-risk features are found in pathology [28]. A comparison of colon cancer screening recommendations is offered in Table 2, with relevant information about the eligible population and type of testing. In the relevant literature column, we cited the source we found for the information from published guidelines, statements, or relevant articles showing the outcomes of the screening methods that are in place.

### 2.2. Breast Cancer

Breast cancer screening depends mostly on imaging techniques such as mammography, ultrasonography, magnetic resonance imaging, and digital breast tomosynthesis, offering a view of the breast and surrounding tissue, including the axillary region. Unfortunately, there is a lack of consensus on stratifying risk groups and offering which method to which type of patient, with a lack of homogeneity observed at both the international and national level [47]. Furthermore, the use of mammography is a subject of dispute among clinicians, due to the risk of irradiation and overdiagnosis (i.e., interpreting benign lesions as malignant, leading to overtreatment, burdening the patient psychologically and financially) [13,48]. Serum biomarkers such as CA15-3 are not suitable for early detection of breast cancer, since the level rarely increases in patients with early or localized breast cancer, although methods have been proposed to improve the sensitivity in early disease detection [49]. Currently, CA15-3 is endorsed for surveillance of resected treated breast cancer or for monitoring progression on treatment for metastatic disease [50].

The USPSTF recommends that all women aged 50–74 years receive every two years screening mammography (grade B), with a suggestion for women aged 40–49 years to receive individualized screening or no mammography [8,51]. However, the ACS recommends that women aged 45–54 receive annual screening mammography and women over the age of 55 annual or biennial mammography until their life expectancy is below 10 years. Women at high risk of breast cancer (such as carriers of BRCA1/2 mutations, women with personal or family history of breast cancer or tumor syndromes such as Li–Fraumeni or Cowden) or with chest radiation therapy administered between 10–30 years of age are recommended to receive breast MRI and yearly mammogram screening starting at age 30 [52]. In Table 3, we summarize the breast cancer screening recommendations in different high-income countries. In the relevant literature column, we cited the source we found for the information from published guidelines, statements, or relevant articles showing the outcomes of the screening methods that are in place.

### 2.3. Prostate Cancer

The recommendations for prostate cancer management include digital rectal examination and serum prostate-specific antigen (PSA) determination for screening. However, evidence so far does not demonstrate a survival improvement for PSA screening. A systematic review of five randomized controlled trials, including a total of 721,718 men enrolled, concluded that testing PSA levels does not affect overall mortality, with a significant risk of overdiagnosis and overtreatment [77]. It is worth remembering that PSA can be detected in the bloodstream after the cancer cells have ruptured the basement layer of the prostate, indicating that the test is conceptually not designed for early tumors or premalignant lesions [78,79].

PSA testing is no longer recommended for prostate cancer screening. Men aged 55–69 have a C grade recommendation according to the USPSTF, meaning that testing is suggested only if recommended by their physician due to increased risk. For men older than 70, the USPSTF recommends against screening. These changes have led to fewer false-positive diagnoses, but also more metastatic prostate cancer diagnoses in the US Caucasian population. Similarly, the ACS recommends discussing PSA testing with men aged 50 and above with an average risk, aged 45 and above who are African American, those with a first-degree relative below the age of 65 diagnosed with prostate cancer, or those aged 40 with multiple first-degree relatives with early PC diagnoses [8,80,81].

### 2.4. Lung Cancer

Low-dose computed tomography (LDCT) is being offered in the United States for high-risk populations (individuals between the ages of 55 and 80 who have a smoking history equivalent to 30 packs of cigarettes per year, who are currently smoking, or have quit within the last 15 years) to screen for lung cancer. LDCT has good results, showing a 20% reduction in lung cancer mortality compared to the conventional chest X-ray, and also higher sensitivity [82]. It is worth noting that in Europe, LDCT does not benefit from the same level of adoption, with several trials failing to show benefit [83,84]. Also, LDCT can give false-positive results, leading to unnecessary diagnostic and therapeutic procedures, as well as false-negative results, as LDCT may not detect all cancers. Radiation exposure and the high cost of this procedure are also limitations for the large-scale adoption of LDCT [82,84,85].

Current recommendations from the USPSTF include annual low-dose computer tomography (LDCT) for adults aged 50–80 with a >20 pack-year history of smoking, and who have either quit within the last 15 years or who are current smokers (grade B recommendation) [86]. LDCT can be discontinued once a patient has not smoked for at least 15 years or if there would not be a benefit on life expectancy any longer. The 2023 ACS Recommendations for Lung Screening follow the USPSTF recommendations, with no limit to the cessation period (all former smokers eligible for LDCT) [87].

### 2.5. Cervical Cancer

Screening for cervical cancer can be performed through a Pap smear followed by cytologic examination of the gynecological tract cells and HPV detection (the causal agent in the overwhelming majority of cases). Pap smears are largely regarded as an example of screening success, due to the impressive reduction in cervical cancer mortality after implementing screening strategies using this method [88], because of the high diagnostic accuracy for premalignant lesions [89].

Cervical cytology is another screening method that benefits from a grade A recommendation from the USPSTF, for women aged 21–29, given every three years, with the choice to continue until 65 years or combine cervical cytology with high-risk human papilloma virus (hrHPV) testing every five years from ages 30–65. The American College of Obstetrics and Gynecology (ACOG) and the Society of Gynecological Oncology (SGO) have both endorsed these recommendations. ACS also adds primary HPV testing as screening every 5 years from age 25–65, with co-testing or cytology (according to USPSTF recommendations) offered only when HPV testing is not available [8,90,91]. Further information on cervical cancer screening recommendations across high-income countries is offered below in Table 4.

### 2.6. Oral Cancer Screening

Visual oral examination (VOE) is the recommended test for oral cancer and oral potentially malignant disorders (OPMDs), which implies systematic visual inspection and palpation of the oral cavity under a bright light source, to detect suspect lesions of the oral mucosa [110,111]. Although most national organizations, including the USPSTF, do not recommend population-based screening through VOE, an RCT showed a reduction in mortality of 81%, and a reduction of 38% in oral cancer incidence [112] and further studies have demonstrated high sensitivity and specificity for this type of screening [110,113]. What is particular about this form of screening is the non-invasiveness, its low cost, and the importance of primary care doctors and dentists, which make the test more accessible and also lowers the burden of manpower needed for population-based screening.

### 2.7. Screening for Other Cancers

Other types of cancers (such as ovarian [114], pancreatic [115], or testicular [116]) may benefit from current methods of screening. However, the current methods have yet to prove a reduction in mortality or reduction in the financial burden, thus lacking a consensus on a standardized valid test. This underlines the need for new validated methods to improve the current management of early detections.

## 3. Fantastic Biomarkers and Where to Find Them

Cancer presence can be discovered in its pre-symptomatic phase through careful evaluation of bodily fluids, where trace amounts of catabolites, nucleic acids, and peptides might reflect the existence of complex mechanisms involved in tumorigenesis and host response directed towards the malignancy. Direct vascular invasion permits cancer cells to be found in blood, but other processes such as necrosis, apoptosis, micro-vesicle budding, and phagocytosis (which occur earlier), or the formation of auto-antibodies, contribute with a host of byproducts that can serve as markers of early-stage disease. As such, bodily fluids can function as a strategic key point of early cancer detection [117,118,119].

### 3.1. Blood Tumor Markers

#### 3.1.1. Circulating Tumor Cells

Circulating tumor cells (CTCs) occur as part of the natural evolution of malignancies towards metastasis. Although this process usually involves lymphovascular invasion, CTCs have been nonetheless found before imaging detection of a primary tumor, indicating a potential value in cancer screening [120,121,122,123].

Huang et al. showed an increase in mean CTC for patients with early (stage I/II) lung, colorectal, gastric, liver, and esophageal cancer, providing a glimpse of establishing a new biomarker, by using EpCAM+/vimentin+ specific immunomagnetic beads for CTC isolation [124,125]. Other possible strategies include the analysis of circulating ensembles of tumor-associated cells (C-ETACs), defined as clusters of pan-cytokeratin and EpCAM-positive cells (at least three). This method has so far outperformed conventional CTC assays, with 89.8% sensitivity and 97% specificity [126,127].

#### 3.1.2. Plasma Cell-Free Nucleic Acids

As described earlier, conventional protein tumor markers such as carcinoembryonic antigen (CEA), cancer antigens 15-3 (CA15-3) or 19-9 (CA19-9), are routinely used in clinical practice, but they suffer from low sensitivity, especially at early stages, and are not routinely used for screening [128]. Novel tests such as liquid biopsies hold promise for accurate screening of multiple tumor types and represent an exciting avenue for the research and development of new biomarkers [129,130,131].

Circulating biomarkers, including microRNAs (miRNAs), cell-free DNA (cfDNA), and circulating tumor DNA (ctDNA), represent active targets as biomarkers of cancer detections and MCED can be achieved through genome sequencing of plasma, as shown by several groups. Malignancy can be detected and tumor sites, including ovary, liver, stomach, pancreas, esophagus, colorectum, lung or breast, can be predicted using nucleic acid assays [17,132].

One of the most discussed tests available is the Galleri (GRAIL) test, which is a targeted methylation assay combined with a machine learning classifier for the detection and discrimination of more than 50 cancers using cfDNA. The test obtained 54.9% sensitivity and 99.3% specificity, with 93% accuracy for tissue-of-origin (TOO) prediction [133,134,135]. Further MCED tests are showing promising results, and we are eagerly awaiting clinical trials to demonstrate their screening value. A review conducted by Brito-Rocha et al. provides further insights into the development of biomarkers that can be evaluated through liquid biopsies, including the clinical trials where these methods were used. The article provides a comprehensive summary of the state of the art in liquid biopsy for MCED research and further encourages the need for thorough clinical research of the new screening tools for them to be used in current practice [18]. As a tribute to their work, we updated Table 3 from their manuscript with the current state of the art in clinical trials for the validation of MCED tests (Table 5 and Table 6).

#### 3.1.3. Extracellular Vesicle-Based Tests for MCED

Extracellular vesicles (EVs), such as exosomes, microvesicles, and apoptotic bodies, are lipid membrane vesicles that get secreted by a large number of cells, including cancer cells. They play major roles in cell–cell-mediated communication, including molecule transportation. Whether they are constituted by nucleic acids, proteins, or metabolites, these molecules can be of interest as cancer biomarkers [136,137,138,139].

Goldvaser et al. studied the use of exosome human telomerase reverse transcriptase (hTERT) mRNA as a minimally invasive pan-cancer biomarker [140]. Their study showed a 62% sensitivity and 100% specificity for detecting 15 cancer types, both solid and hematologic.

The Verita^TM^ test using EV-protein profiling combined with machine learning is capable of detecting pancreatic, ovarian, and bladder cancer in early, curative stages (stage I and II), with varying degrees of sensitivity pancreatic cancer (95.5%), ovarian cancer (74.4%), and bladder cancer (43.8%) [141]. Most encouragingly, targeting the CD9+ EVs and aptamer recognition of CD63/EpCAM/MUC1 (epithelial markers), carcinomas were detected with 100% sensitivity and specificity, indicating that specific targeting of tumor type-specific proteins can be used for both cancer screening and identification of the tumor’s origin [142,143].

### 3.2. Non-Invasive Early Cancer Detection Using Indirect Media—Saliva, Tears, Urine, Breath

Blood is not the only compartment where biomarkers can be identified. Other media, for instance, urine, saliva, or breath, may contain materials derived from circulation. These molecules can be interrogated through non-invasive testing for the oncogenic pathways they uncover [17].

Although the glomerular filter impedes the passing of larger molecules, extracellular vesicles containing tumor-specific RNA signatures can be detected through a nanowire device embedded in a microfluidic system, as described by Takao Yasui et al. [144].

Furthermore, saliva is an easily available bodily fluid that may emerge as a haven for MCED development, as innovations in the assay of circulatory biomarkers have been exciting thus far. The review by Kaczor-Urbanowicz et al. delves deeper into the recent progress made in salivaomics and the impact on cancer diagnosis [145].

Tear samples, although difficult to collect, maybe an interesting avenue for MCED biomarker discovery, as a study by Daily et al. has identified several protein markers in tears of early-stage breast cancer [146].

Last but not least, exhaled breath can become a viable site for biomarker identification, with devices showing promising results in detecting multiple cancers, including respiratory, gastrointestinal, or gynecological cancers. We recommend the systematic review done by Krilaviciute et al. for further insights on exhaled breath biomarker discovery [147].

### 3.3. Tumor Exfoliation

Anatomic corridors such as the gastrointestinal (GI) tract or the female reproductive tracts are lined by epithelial cells, which have a physiological tendency to exfoliate into a common efflux route. Similar to CTCs, the excretory medium for these exfoliates can be interrogated for clinically useful biomarkers, as exfoliation occurs from both precursor lesions and earliest-stage cancers, even before any substantial invasion can occur [17].

The stool can be a valuable sample for colorectal cancer screening, since even precursor lesions can be accurately detected using multi-target DNA tests, achieving detection rates comparable to those of colonoscopy [148,149,150]. Attempts at improving diagnostic accuracy have been made by targeting molecular debris in stool, including abnormal DNA (such as mutant KRAS, actin, FIT, aberrantly methylated BMP, and NDRG4 promoter regions), using the Cologuard^®^ stool DNA test. Unfortunately, the test had lower sensitivity towards adenomas, thus limiting its preventive role, and lower specificity compared to FIT (87–90% compared to 95–96%) [17,150]. The mt-sRNA ColoSense^®^ test showed improvements in sensitivity (94%), but remains insufficient for advanced adenomas (46%) [151]. However, molecular tumor fingerprinting of early cancers using stool samples remains an attractive venue in biomarker research. Also, compared to the standard fecal occult blood test, early data suggest that stool testing can be expanded to include the previously unscreened upper GI tract as well. However, techniques need to be optimized and rigorously tested in the clinic to be adequately assessed to have a pan-GI screening test [17]. In our opinion, the diagnostic yield of these tests is dependent on accurately predicting the mutations occurring earliest in tumorigenesis, which may influence the transition between premalignant lesions towards neoplasia [152].

Detecting gynecological neoplasms with tampons is a concept that has been proven by analyzing methylated DNA markers extracted from vaginal tampons. Also, molecular analysis may allow the detection of both endometrial and ovarian cancer using a conventional cervical Papanicolaou test, further expanding the clinical utility of this routine test. Further clinical trials are needed to demonstrate the practicality of these tests, as they might turn a single cancer detection test into an MCED examination [153,154].

**Table 5 cancers-16-01191-t005:** Clinical trials conducted/ongoing for validation of multi-cancer early detection tests (updated and redesigned from Table 3 of Brito-Rocha et al. [18]).

Trial ID	Trial Name	MCED Test	Sponsor	Status (as of February 2024)	Est. Completion	Enrolled	Publications
**NCT02889978**	CCGA, GRAIL-001	Galleri	GRAIL, LLC	Active, not recruiting	March 2024	15,254	[135,155]
**NCT03085888**	STRIVE, GRAIL-002	Galleri	GRAIL, LLC	Active, not recruiting	May 2025	99,481	
**NCT03934866**	SUMMIT	Galleri (combined with LDCT)	University College London and Grail	Active, not recruiting	August 2030	13,035	[156]
**NCT04241796**	PATHFINDER, GRAIL-007	Galleri	GRAIL, LLC	Completed	NA	6662	[157,158,159]
**NCT05155605**	PATHFINDER2	Galleri	GRAIL, LLC	Recruiting	January 2028	35,000	
**NCT05205967**	REFLECTION	Galleri	GRAIL, LLC	Recruiting	August 2026	17,000	
**NCT05611632,** **ISRCTN91431511**	NHS-Galleri, GRAIL-009	Galleri	GRAIL, Bio UK Ltd.	Active, not recruiting	February 2026	140,000	[160,161,162]
**NCT05235009**	LEV87A	GAGome	Elypta	Recruiting	March 2025	9170	[163]
**NCT05295017**	LEV93A	GAGome	Elypta	Recruiting	March 2025	1256	
**NCT05780957**	Multi-Cancer Early Detection (MCED) of Firefighters	GAGome	Elypta	Recruiting	January 2031	2000 (estimated)	
**NCT05227534**	PREVENT	OverC	Burning Rock Dx	Recruiting	January 2029	12,500	-
**NCT04972201**	PROMISE	OverC	Chinese Academy of Medical Sciences and Burning Rock Dx	Completed	NA	2305	[164]
**NCT04822792**	PRESCIENT	OverC	Chinese Academy of Medical Sciences and Burning Rock Dx	Completed	NA	11,879	-
**NCT04820868**	THUNDER	OverC	Shanghai Zhongshan Hospital and Burning Rock Dx	Completed	NA	2508	[165,166]
**NCT04817306**	PREDICT	OverC	Shanghai Zhongshan Hospital and Burning Rock Dx	Completed	NA	14,206	-
**NCT03517332**	-	DEEPGEN	Quantgene Inc.	Completed	NA	675	[167]
**NCT03967652**	-	Na-nose	Anhui Medical University	-	NA	NA	[168,169]
**NCT05366881**	CAMPERR	-	Adela, Inc.	Active, recruiting	December 2026	7000	[170]
**NCT05254834**	Vallania	-	Freenome Holdings Inc.	Active, recruiting	June 2025	5400	-
**NCT05227261**	K-DETEK	SPOT-MAS	Gene Solutions	Active, recruiting	-	10,000	[171,172]
**NCT05159544**	FuSion	-	Singlera Genomics Inc.	-	-	60,000	[173]
**NCT04405557**	PREDICT	-	Geneplus-Beijing Co.	Completed	NA	757	Article under review
**NCT02662621**	EXODIAG	-	Centre Georges Francois Leclerc	Completed	NA	80	[174]
**NCT04197414**	-	-	Yonsei University	Active, recruiting	December 2029	3000	-
**NCT03951428**	-	-	LifeStory Health Inc.	Completed	NA	30	-
**NCT03869814**	-	-	Bluestar Genomics Inc.	Completed	NA	660	-
**NCT02612350**	-	-	Pathway Genomics	Completed	NA	1106	-
**NCT06011694**	The Jinling Cohort	MERCURY test	Nanjing Shihejiyin Technology, Inc.	Recruiting	May 2027	15,000 (estimated)	[175]
**NCT05633342**	CADENCE	MiRXES MCST	MiRXES Pte Ltd.	Recruiting	May 2025	15,000 (estimated)	-
**NCT04825834**	DELFI-L101	DELFI	Delfi Diagnostics Inc.	Recruiting	June 2026	2660 (estimated)	[176]
**NCT05306288**	DELFI-L201 (CASCADE-LUNG)		Delfi Diagnostics Inc.	Active, not recruiting	March 2025	15,000 (estimated)	-
**NCT05492617**	DETECT		Rigshospitalet, Denmark	Completed	NA	3628	[177]
**NCT05435066**	CORE-HH	Harbringer Hx	Harbringer Health	Recruiting	July 2025	10,000 (estimated)	[178]
**NCT05117840**	SHIELD	LUNAR-2	Guardant Health, Inc.	Recruiting	January 2026	9000 (estimated)	[179,180]
**NCT04136002**	ECLIPSE				January 2024	40,000	[181]

**Table 6 cancers-16-01191-t006:** Technical description of MCED Tests currently on clinical trials conducted/ongoing for validation of multi-cancer early detection tests (updated and redesigned from Table 3 of Brito-Rocha et al. [18]).

MCED Test	Sponsor	Publications	Sample	Method of Detection	Tumor Types
Galleri	GRAIL, LLC	[135,155]	plasma	Targeted methylation assay (cfDNA), combined with a machine learning classifier for detecting (54.9% sensitivity, 99.3% specificity) and discriminating TOO (93% accuracy) [134,135].	<40 different tumor types: GI, hematological, head and neck, lung, ovary, etc.
GAGome	Elypta	[163]	plasma and urine	Free glycosaminoglycan profiles (GAGome)	14 different cancers: gliomas, genito-urinary, etc.
OverC	Burning Rock Dx	[164]	plasma	Targeted methylation sequencing assay (ELSA-seq) combined with machine learning for cancer detection (80.6% sensitivity, 98.3% specificity) and TOO discrimination (81.0% accuracy)	Lung, colorectal, liver, ovarian, pancreatic, esophageal [182]
DEEPGEN	Quantgene Inc.	[167]	Plasma	NGS combined with machine learning for cancer detection (57% sensitivity, 95% specificity)	Lung, breast, colorectal, prostate, bladder, pancreatic, and liver
Na-nose	Anhui Medical University	[168,169]	Exhaled breath	Nanoparticle sensors for Volatile Organic Compounds found in exhaled breath	Head and neck, lung, breast, colorectal, etc.
-	Adela, Inc.	[170]	Plasma	cfMeDIP-seq [183] combined with machine learning	Kidney, pancreas, lung, gliomas, etc.
SPOT-MAS	Gene Solutions	[171,172]	Plasma	NGS sequencing of ctDNA to detect genome-wide hypomethylation, targeted hypermethylated regions, genome-wide copy number variation, length profile of ctDNA	Gastric, lung, breast, etc.
-	Singlera Genomics Inc.	[173]	Plasma	DNA methylation	lung, esophagus, stomach, liver, pancreatic and colorectal
-	Geneplus-Beijing Co.	Article under review	Plasma	cfDNA analysis	Hepatic, pancreatic, ovarian, breast, colorectal and gastric cancer.
-	Centre Georges Francois Leclerc	[174]	Blood and urine	Analysis of HSP70-exosomes	Breast, lung, ovarian
-	Yonsei University	-	Blood and urine	ctDNA analysis	Urological malignancies
-	LifeStory Health Inc.	-	Menstrual blood		Breast, endometrial, lung
-	Bluestar Genomics Inc.	-	Plasma	cfDNA analysis	Pancreatic
-	Pathway Genomics	-	Plasma	cfDNA analysis	Pancancer
MERCURY test	Nanjing Shihejiyin Technology, Inc.	[175]	plasma	cfDNA analysis	Liver, lung, colorectal
MiRXES MCST	MiRXES Pte Ltd.	-	plasma	miRNA expression in tandem with DNA methylation	lung, breast, colorectal, liver, stomach, esophageal, ovarian, pancreatic, and prostate
DELFI	Delfi Diagnostics Inc.	[176]	plasma	cfDNA analysis	Lung, liver cancer
Harbringer Hx	Harbringer Health	[178]	plasma	cfDNA analysis	Variety of solid and hematological cancers
LUNAR-2	Guardant Health, Inc.	[179,180]	plasma	ctDNA analysis	Colon, lung cancer

## 4. Expanding the Horizons towards Universal Cancer Screening

### 4.1. Improved Imaging Techniques

Conceptually, computed tomography (CT) scans with or without contrast enhancement of the whole body can be considered MCED. However, CT scans have low sensitivity and specificity, and pose a significant threat of side effects, due to irradiation and contrast-induced nephropathy, which severely limit the repeated use for one person [184,185].

Next-generation approaches towards imaging involve molecular, nanoparticle, and fluorescent constructs, which have sparked clinical interest for their potential to capture a precise in vivo snapshot of the person’s oncological status, providing details for both screening and treatment monitoring. Further research is warranted to optimize the technical details and to provide information about the examination performance and clinical outcomes [17,186,187,188].

### 4.2. Volatile Organic Compounds

Byproducts of metabolic processes involving cellular transformation and tumorigenesis can be found in trace amounts in urine and expired breath. These catabolites and the patterns of volatile organic compounds (VOCs) can be studied, with numerous studies indicating potential relationships between VOCs and different cancers. It has been suggested that in breast cancer patients, six specific VOCs could be used for a screening test. Other studies have highlighted the importance of VOCs in cancers such as prostate, kidney, or lung [7,133,169,189,190,191,192,193,194,195,196,197].

So far, volatilome research has been impeded by technical issues and a high financial burden, requiring complex technology for the identification of specific patterns. However, a biosensor has recently been proposed, *Caenorhabditis elegans*, a nematode with a refined olfactory sense (with over 1200 olfactory receptor-like genes, encoding 1.5 times as many different types of olfactory receptors as a dog, capable of detecting subtle odorant concentration changes in its habitat). With the observation that *C. elegans* is attracted to urine from cancer patients, but avoids urine from healthy persons, a team led by Hirotsu et al. perfected an innovative, highly sensitive cancer screening test called N-Nose, which has been available in Japan since 2020. It is capable of detecting the presence of cancer for 15 malignancies, with a sensitivity of 87.5% and specificity of 90.2%. Unfortunately, N-Nose only correlates with the presence/absence of cancer, without the capability of pinpointing the cancer types. However, efforts are being made to further understand the chemotaxis mechanism of *C. elegans* and methods of perfecting this potentially powerful screening tool. It is important to mention that the N-Nose behavioral assay has been thoroughly evaluated in clinical studies with large cohorts of patients, showing reliable results. Also, the results have been replicated at other prestigious institutions in the US and Italy [7,128,198,199,200,201,202,203].

### 4.3. Always on PATROL for Early Lung Cancer

A particular assay that has drawn the authors’ attention has been the diagnostic platform PATROL (point-of-care aerosolizable nanosensors with tumor-responsive oligonucleotide barcodes), which is currently tested for the early diagnosis of lung cancer, but with the potential to become an MCED test for airway cancers. The diagnostic platform incorporates three modules, as follows. (1) Artificially synthesized low-plex activity-based nanomers (ABNs), consisting of protease substrates (for dysregulated proteases produced by tumoral cells on the lung epithelia) conjugated to eight-armed PEG nanoscaffolds with single-stranded modified DNA 20-mers. These molecules are nebulized through a (2) portable inhalation unit, and then delivered to the lung, where the DNA barcodes are liberated upon substrate cleavage by dysregulated proteases associated with early lung adenocarcinoma, shed into the systemic circulation, and then excreted by the kidneys in the urine. The urine is then sampled (2 h after inhalation), and the DNA barcodes are analyzed through a (3) multiplex lateral flow assay, quantified, and classified to provide diagnosis. The results of this test have so far been promising with high sensitivity and specificity for stage I/II lung adenocarcinoma [204]. This particular assay sparked our interest not only for the good results shown so far and the cost-effectiveness promised by the authors but also for the concept of activity-based diagnostics (ABD), in which the tumoral metabolic processes are leveraged to provide the result via the excretion of a synthesized biomarker. We believe that variants of this concept can be proposed and used for other cancers by mastering the triad this assay proposes: (1) a highly sensible, conjugable complex consisting of a readable DNA barcode and a cleavable enzyme substrate that can be targeted by dysregulated tumoral enzymes; (2) an accessible, preferably non-invasive entry route for this complex, which cannot be degraded along this path, towards the organ(s) in which cancer is screened; and (3) an excretion route where the DNA can be available for analyzing and interpretation of the results.

## 5. Discussion on the Prospect of Universal Cancer Screening

### 5.1. “One Test to Rule Them All” vs. “One Organ at a Time”

The universal cancer screening approach is a paradigm-changing approach for the early detection of cancers through the non-invasive interrogation of multiple markers and media on the possible existence of malignancy. With the potential of multi-organ detection, we can imagine tests that can include all cancers in a single efficient modality of testing, which would be potentially both cost- and life-saving. Perhaps, in the era of treating cancer using “precision medicine” tailored to the most intimate characteristics of tumor cell biology, a “one test to rule them all” approach toward cancer screening would be a match made in heaven, as simulations have already modeled, promising reductions of up to a quarter in all cancer-related deaths in patients aged 50–79 years in the United States and a reduction of almost 80% for late-stage (III, IV) disease [7,17,205].

### 5.2. Aggregate Prevalence

The value of a screening intervention is given by the combination of a low estimated number of persons that need to be screened to detect one cancer (NNS), and a high positive predictive value (PPV—the probability that a screening test indicates the presence of a tumor). Given that prevalence varies across tumor locations and subtypes, a single test that incorporates multiple targets can decrease the NNS. Furthermore, aggregating multiple cancers with different prevalence rates in a universal test would yield higher PPVs than single-organ screening. This argument has been comprehensively supported by Ahlquist et. al, offering an insightful rationale for the need for MCED [17].

### 5.3. Manpower Considerations

It is difficult to estimate if MCED screening would represent a burden or a relief for medical healthcare workers. One of the pivotal requirements of a successful, efficient screening program is a central, well-funded authority, responsible for setting recommendations and organizing the framework of how these recommendations can be applied (see Table 2, Table 3 and Table 4). It is most encouraging to see cancer screening programs offering eligible population tests that do not require healthcare workers (such as FIT) at home [35]. These tests are generally accompanied by greater compliance from the general public because of their non-invasiveness, privacy, and simple nature (see Table 1). Collecting samples such as stool, urine, or saliva can be done by the patient and sent through the mail to a specialized laboratory. Though most of the MCED screening tests in clinical trials (see Table 5 and Table 6) require blood samples, this should not be an impediment, as these tests can be offered simultaneously with other regular health checkups that require this minimally invasive medical procedure. In conclusion, we believe that MCED would not burden already-pressured health services.

### 5.4. Cost-Effectiveness

Most cancers are diagnosed in late stages, needing expensive long-term systemic treatment, which leads to an increase in the economic burden of cancers. In 2020 alone, the US healthcare system covered 157.7 billion USD, and 16.1 million cancer patients had out-of-pocket expenses, with a quarter of them declaring they had difficulties paying the bills. Mitigation of the economic, medical, and psychological issues related to late-stage diagnosis is an overarching drive of universal cancer screening. Such expenses warrant investments in MCED, as tests such as Galleri (~945 USD), or N-Nose (~109 USD) can be cheaper than PET-CT, for instance (~1100 USD). Large-scale implementation of these tests can lower the price even further, and the biggest benefit would be offering prompt treatment, with lower financial burden for both the patient and healthcare providers [7,206]. The reductions in the financial burden have been estimated by Hackshaw et al. and Tafazzoli et al., showing that hypothetically MCED could reduce the cost per cancer treatment by up to 5421 USD per cancer patient [19,207]. With permission, we decided to present the estimated data from Hackshaw et al.’s article in Table 7.

### 5.5. Bioinformatics and Artificial Intelligence (AI)

Biomarker research has evolved in close relation with bioinformatic development. Since the development of the Cancer Genome Atlas (TCGA) project, large databases containing valuable data on the multisystem biology features of multiple tumor types have led to the development of data mining methods to propose valuable biomarkers [18,208,209,210,211,212,213,214]. Examples include the study of molecular aberrations of the SST gene for multiple GI cancers [215], the difference in HSP90α protein across 9 different tumors [216], or the identification of CpG markers to detect 27 cancer types [217].

Furthermore, the existence of large databases allows machine learning algorithms to be developed and trained to improve the detection capacity of current biomarkers. Several neural network-based algorithms have been developed to improve detection sensitivity and specificity, as well as tumor-of-origin (TOO) discrimination [18]. The undeniable benefit of artificial intelligence becomes evident when we observe that the vast majority of MCED tests with ongoing trials have applied deep learning algorithms to improve their diagnostic yield (Table 6). As such, it is important to recognize AI as a key element of biomarker discovery that boosts the output and outcomes of MCED tests.

## 6. Conclusions

The dream of universal cancer screening remains elusive at the moment. However, translational studies coupled with clinical research offer us an optimistic view for the future, as we move closer to significant improvements in oncological outcomes through early diagnosis and curative treatment. We are hopeful that the tools presented in this article will soon be incorporated into current guidelines for screening the general population and will help improve the survival rate of cancer patients.

## Figures and Tables

**Figure 1 cancers-16-01191-f001:**
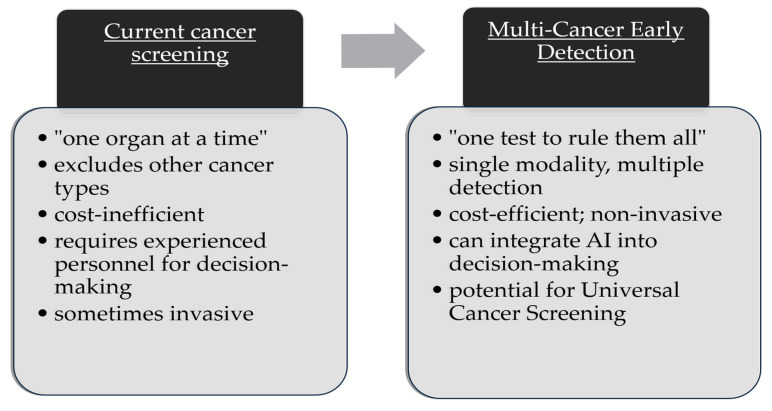
Comparison between the present concept of screening versus multi-cancer early detection screening. Inspired by Ahlquist et. al, and redesigned [17].

**Table 1 cancers-16-01191-t001:** Differences in screening recommendations and adherence for the United States and the United Kingdom, with data regarding adherence, sensitivity, and specificity (data from Supplementary Table S1 from Hackshaw et al. [19]).

Cancer	Screening Procedure		Screening Interval		% Eligible Adults		Adherence		Sensitivity		Specificity	
	US	UK	US	UK	US	UK	US	UK	US	UK	US	UK
Lung	Low-dose computed tomography	No Recommendation	Annually	NA	7%	NA	14.4%	NA	84.9%	NA	87.2%	NA
Colorectal	Cologuard	FIT	Every 3 years	Every 2 years	93%	68%	67.7%	57.7%	92%	79%	87%	94%
Breast	Bilateral mammography		Annually	Every 3 years	90% of women	77% of women	77.6%	70.5%	87%	87%	89%	97.2%
Cervical	Cytological examination and hrHPV testing		Every 5 years	Every 5 years	63% of women	60% of women	95%	76.2%	95%	95%	85%	85%

**Table 2 cancers-16-01191-t002:** Comparison between colon cancer screening recommendations for the average-risk population in different high-income countries (inspired by Ebell et al. [29], and updated).

Country	Organization	Type	Year	Test	Eligible Population	Relevant Literature
United States	USPSTF	National Guideline Committee	2021	Shared decision-making between clinician and patient for one of the following strategies: - g-FOBT/FIT every year. - Stool DNA-FIT every 1–3 years. - CT colonography every 5 years. - Flexible sigmoidoscopy every 5 years. - Flexible sigmoidoscopy every 10 years and annual FIT. - Colonoscopy screening every 10 years.	All adults aged 45–75 years; selectively screen adults aged 76–85 years	[27]
	ACS	Cancer Society	2018	- g-FOBT/FIT every year. - Stool DNA-FIT every 3 years. - Colonoscopy every 10 years - CT colonography every 5 years. - Flexible sigmoidoscopy every 5 years.	All adults aged 45–75; selectively screen adults aged 76–85 years	[28]
Luxembourg	Ministry of Health	National Guideline Committee	2016	FIT every 2 years. Colonoscopy if positive test.	All adults between 55–74 years old invited to participate	NA
Switzerland	League Against Cancer	Cancer Society	2013	Colonoscopy every 10 years—g-FOBT every two years.	All adults aged 50–69	[30]
Norway	Cancer Registry of Norway	Cancer Society	2022	FIT every 2 years. Colonoscopy if positive test.	All adults aged 55–74	[31]
Netherlands	NIPHE	National Guideline Committee	2019	FIT every 2 years.	All adults aged 55–75	[32]
Germany	Federal Joint Committee	National Guideline Committee	2019	Women, 50–54: annual FIT. Women >55: FIT every 2 years or colonoscopy twice at an interval of 10 years.	All adults from 55 years onwards	[33]
				Men, 50–54: annual FIT or colonoscopy twice at an interval of 10 years. Men, >55: FIT once every 2 years or colonoscopy twice at intervals of 10 years.		
Sweden	National Board of Health and Welfare	National Guideline Committee		FIT once every 2 years. Colonoscopy if positive test.	All adults aged 60–74	[34]
Ireland	National Screening Service	National Guideline Committee	2013	Bowel Screen Programme: FIT once every 2 years. Colonoscopy if positive test.	All adults aged 59–69	[35]
Austria	Austrian Cancer Care	Cancer Society	2023	Colonoscopy every 10 years, or FIT every 2 years.	All adults 45–75 years	[36]
Denmark	National Board of Health	National Guideline Committee	2014	FIT once every 2 years. Colonoscopy if positive test.	All adults aged 50–74	[37]
Belgium	Foundation Against Cancer	Cancer Society	2013	FIT once every 2 years. Colonoscopy if positive test.	All adults aged 50–74	[38]
Canada	CTFPHC	National Guideline Committee	2016	50–59: g-FOBT/FIT every 2 years or flexible sigmoidoscopy every 10 years (weak recommendation). 60–74 years: g-FOBT/FIT every 2 years or flexible sigmoidoscopy every 10 years (strong recommendation).	All adults aged 50–74 years	[39]
Australia	Australian Government Department of Health	National Guideline Committee	2023	FIT once every 2 years. Colonoscopy if positive test.	All adults aged 50–74 years	[40]
France	National Cancer Institute	National Guideline Committee	2015	FIT once every 2 years. Colonoscopy if positive test.	All adults aged 50–74 years	[41]
Japan	National Cancer Center	National Guideline Committee	2016	g-FOBT once per year.	All adults aged 40 and over	[42]
Iceland	Icelandic Cancer Society	Cancer Society	2015	FIT once every 2 years. Colonoscopy if positive test.	All adults aged 60–69	NA
UK	UK National Screening Committee	National Guideline Committee	2020	FIT once every 2 years. Colonoscopy if positive test.	All adults aged 50–74	[43]
Finland	Cancer Society of Finland	Cancer Society	2021	FIT once every 2 years. Colonoscopy if positive test.	Adults aged 56–74 years old	NA
New Zealand	Ministry of Health	Cancer Society	2023	Time to Screen program offering FIT once every 2 years. Colonoscopy if the test is positive	Adults aged 60–74	[44]
Italy	National Screening Observatory	National Guideline Committee	2015	FIT once every 2 years. Colonoscopy if the test is positive.	Adults aged 50–69 years	[45]
Spain	Cancer Strategy of National Health System	National Guideline Committee	2009	FIT once every 2 years.	Adults aged 50–69	[46]

Abbreviations: CTFPHC = Canadian Task Force on Preventive Health Care; NA = Not available; NIPHE = National Institute for Public Health and the Environment, USPSTF = United Stated Preventive Services Task Force.

**Table 3 cancers-16-01191-t003:** Comparison between breast cancer screening recommendations in different high-income countries (inspired by Ebell et al. [29], and updated).

Country	Organization	Type	Year	Primary Group	Further Discussion	Relevant Literature
United States	USPSTF	National Guideline Committee	2023 *	Women aged 40–75: biennial screening mammography	Insufficient evidence for mammography screening in women >75	[53]
					Insufficient evidence for breast US, or MRI in women identified with dense breasts on an otherwise negative screening mammogram	
	ACS	Cancer Society	2015, 2007	Women with average risk of breast cancer: ages 40–45—option to start yearly mammography ages 45–54—yearly mammography; ages ≥55—biennial/annual screening; continue mammography until life expectancy ≥10 years	Annual breast MRI screening in higher-risk patients: BRCA mutation, a first-degree relative of BRCA carrier (untested), the lifetime risk of ≥20–25% (as defined by BRCAPRO/other models), radiation to the chest between 10–30 years, Li–Fraumeni/Cowden/Bannayan–Riley–-Ruvalcaba syndrome and first-degree relatives	[52,54]
	ACOG	Specialty Society	2017, reaffirmed 2021	Offer mammography starting at 40 years. Initiate at ages 40–49 years after counseling. Recommend by no later than age 50 if the patient has not already been initiated. Continue until age 75. Beyond 75, consider life expectancy.	Annual or biennial screening. Clinical breast examination may be offered for women aged 25–39 years and annually for women 40 years and older.	[55]
	ACR	Specialty Society	2023, 2021	Risk assessment by age 25. Annual mammography starting at the age of 40. Screening should continue past age 74 unless severe comorbidities limit life expectancy.	Women with genetically based increased risk, or with a calculated lifetime risk of ≥20% are recommended to undergo breast MRI surveillance starting at ages 25–30 and annual mammography starting between ages 25–40, depending on the risk.	[56,57]
Luxembourg	Ministry of Health	National Guideline Committee	NA	Programme Mammographie targets women aged 50–70, inviting them for mammography every 23 months.		
Switzerland	Swiss Cancer League	Cancer Society	2016	Mammography once every 2 years for women aged 50–70 (74 in some cantons)		[58]
Norway	Cancer Registry of Norway	Cancer Society	2010	Mammography once every 2 years for women 50–69 years of age		[59]
Netherlands	NIPHE	National Guideline Committee	2017	Mammography once every 2 years for women 50–74 years of age		[60]
Germany	Federal Joint Committee	National Guideline Committee	2015	Mammography once every 2 years for women 50–69 years of age		[61]
Sweden	National Board of Health and Welfare	National Guideline Committee	2013	Mammography once every 2 years for women 40–74 years of age		[62]
Ireland	National Screening Service	National Guideline Committee	2015	BreastCheck program offers mammography once every 2 years for all women aged 50–69 years.		[63]
Austria	Austrian Cancer Aid Society	Cancer Society	2014	Mammography once every 2 years for women aged 45–69 years		[64]
Denmark	National Board of Health	National Guideline Committee	2014	Mammography once every 2 years for women aged 50–69 years		[65]
Belgium	Foundation Against Cancer	Cancer Society	2017	Mammography once every 2 years for women aged 50–69 years		[66]
Canada	CTFPHC	National Guideline Committee	2011	Mammography once every 2–3 years for participants aged 50–74	For women aged 40–49 years, recommendation against mammography (shared decision-making with healthcare provider)	[67]
Australia	Australian Government Department of Health	National Guideline Committee	2015	BreastScreen program offers mammography every 2 years for women aged 50–74	Program available for women from age 40, with mammography every 2 years	[68,69]
France	National Cancer Institute	National Guideline Committee	2015, 2016	Mammography once every 2 years for participants aged 50–74. Screening also includes manual breast exam	Yearly MRI, mammography, and US for patients from the age of 30 onwards in high-risk cases (genetic factor, family history, personal history of radiation treatments)	[70]
Japan	National Cancer Center	National Guideline Committee	2021	Mammography once every two years to women aged 40–69 ± manual breast examination		[71]
Iceland	Icelandic Cancer Society	Cancer Society	NA	Mammography once every 2 years for women aged 40–69 years. Invitation for women aged 70–74 to have mammography every three years		[72]
UK	UK National Screening Committee	National Guideline Committee	2012	Mammography once every 3 years form women aged 50–70 years		[73]
Finland	Cancer Society of Finland	Cancer Society	2010	Mammography once every 2 years for women aged 50–69 years		[72]
New Zealand	Ministry of Health	Cancer Society	2014	BreastScreen Aotearoa program offers mammography every two years for women aged 45–69		[74]
Italy	National Screening Observatory	National Guideline Committee	2015	Mammography once every 2 years for women aged 50–69 years (45–69 in Campania region)		[75]
Spain	Cancer Strategy of National Health System	National Guideline Committee	2009	Mammography once every 2 years for women aged 50–69 years		[76]

* Currently a draft recommendation at the time of writing (March 2024). Abbreviations: ACOG = American College of Obstetrics and Gynecology, ACR = American College of Radiology; ACS = American Cancer Society; CTFPHC = Canadian Task Force on Preventive Health Care; NA = Not Available; NIPHE = National Institute for Public Health and the Environment, USPSTF = United Stated Preventive Services Task Force.

**Table 4 cancers-16-01191-t004:** Comparison between cervical cancer screening recommendations in different high-income countries (inspired by Ebell et al. [29], and updated).

Country	Organization	Type	Year	Screening Method	Eligible Population	Relevant Literature
United States	USPSTF	National Guideline Committee	2018	Women aged 21–29: cytology alone every 3 years Women aged 30–65 years: cytology alone every 3 years, with HR-HPV testing alone every 5 years, or co-testing every 5 years	Women between years 21–65	[91]
	ACS	Cancer Society	2020	Screen with HR-HPV testing every 5 years If not available, co-testing every 5 years, or cytology every 3 years	Women between years 25–65	[90]
	ACOG	Specialty Society	2021	Endorses USPSTF recommendations	Women between years 21–65	[92]
Luxembourg	Ministry of Health	National Guideline Committee	NA	NA	NA	NA
Switzerland	League Against Cancer	Cancer Society	2018	Women aged 21–29: cytological screening Women aged 30–70 years: cytological/HR-HPV screening every 3 years	Women between years 21–70	[93]
Norway	Cancer Registry of Norway	Cancer Society	2022	Cytology screening from the age of 25, HR-HPV testing from the age of 34	Women aged 25–69 years	[94]
Netherlands	NIPHE	National Guideline Committee	2021	HR-HPV screening every five years between 30–60 years	Women aged 30–60	[60]
Germany	Federal Joint Committee	National Guideline Committee	2020	Women aged 21–35: annual genital examination, medical history, cytology Women aged >35 are offered HR-HPV and cytology co-testing	Women aged 20	[95,96]
Sweden	National Board of Health and Welfare	National Guideline Committee	2015	Women aged 23–29: Cytology-based screening once every three years Women aged 30–49: HR-HPV testing every three years with a cytology co-test at age 41 Women aged 50–64: HR-HPV test every seven years	Women aged 23–64	[97]
Ireland	National Screening Service	National Guideline Committee	2020	CervicalCheck program—women aged 25–29 years: cervical screening every 3 years Women aged 30–65 years: cervical screening every 5 years	Women between 25–65 years of age	[98]
Austria	Austrian Cancer Care	Cancer Society	2014	Cytology every 3 years	Women between 19–69 years of age	[99]
Denmark	National Board of Health	National Guideline Committee	2022	Women aged 23–49 years: co-testing every 3 years Women aged 50–64 years: co-testing every 5 years	Women between 23–64 years of age	[100,101]
Belgium	Foundation Against Cancer	Cancer Society	2022	Cytology tests every 3 years Women aged 30–64 years—HR-HPV test every five years	Women aged between 25–64 years	[102]
Canada	CTFPHC	National Guideline Committee	2013	Cytology every 3 years	Women aged between 25–69 years	[103]
Australia	Australian Government Department of Health	National Guideline Committee	2022	HR-HPV testing every 5 years with cytology for positive tests	Women aged 25–74 years of age	[104]
France	National Cancer Institute	National Guideline Committee	2022	Women 25–30 years: cytology testing every 3 years Women 30–65 years HR-HPV testing every 5 years	Women aged 25–65 years	[105]
Japan	National Cancer Center	National Guideline Committee	2020	Women aged 20–69 years: cytology testing once every 2 years	Women aged 20–69	[106]
Iceland	Icelandic Cancer Society	Cancer Society	2020	Women aged 23–29 years: HR-HPV every three years Women aged 30–59 years: HR-HPV every five years Women aged 60–64 years: HR-HPV test—if a test is negative, women are discharged from screening	Women aged 23–64	NA
UK	UK National Screening Committee	National Guideline Committee	2019	HR-HPV test once every 3 years	Women aged 25–64	[107]
Finland	Cancer Society of Finland	Cancer Society	2017	Cytology once every five years (can be substituted with HR-HPV)	Women aged 25–65	[108]
New Zealand	Ministry of Health	Cancer Society	2023	HR-HPV once every five years (once every 3 years if immunodeficient) screening extended to women aged 70–74 years if unscreened or under-screened	Women aged 25–69	[104]
Italy	National Screening Observatory	National Guideline Committee		Women aged 25–30 years: Cytology test once every three years Women aged 30–65 years: HPV-based screening every five years	Women aged 25–65 years	[97]
Spain	Cancer Strategy of National Health System	National Guideline Committee	2019	Women aged 25–34 years: cytology co-testing once every 3 years Women aged 35–65 years: HR-HPV testing every five years	Women aged 25–65 years	[109]

Abbreviations: ACOG = American College of Obstetricians and Gynecologists; ACS = American Cancer Society; CTFPHC = Canadian Task Force on Preventive Health Care; NA = Not Available; NIPHE = National Institute for Public Health and the Environment.

**Table 7 cancers-16-01191-t007:** Estimates of screening outcomes for the UK, per year, from Hackshaw et al. [19] with permission from the authors.

	Current Screening Only						Incremental MCED Test			
Analysis	Total Positives	True Positive Cancers (Diagnostic Yield per 1000)	True Positive/False Positive Ration	Cancer Detection Rate	Diagnostic cost per Confirmed Cancer Diagnosis	Total Positives	Diagnostic Yield per 1000	True Positive/False Positive Ration	Cancer Detection Rate	Diagnostic Cost per Confirmed Cancer Diagnosis
100% MCED uptake	481,876	24,888 (3.43)	1:18	12%	10,452	244,153	92,817 (4.26)	1:1.6	43%	2175
20% decrease cancer incidence in screening population	476,899	19,910 (2.74)	1:23	9%	12,921	246,220	94,884 (4.35)	1:1.6	44%	2134
Difference	−4977	−4978	-	-	2469	2067	2067	-	-	−41
Prostate cancer added	635,797	27,754 (2.61%)	1:22	13%	12,917	243,586	92,250 (4.23%)	1:1.6	43%	2187
Difference	153,921	2866			−2465	−567	−567	-	-	12
50% MCED uptake	-	-	-	-	-	122,076	46,409 (4.23)	1:1.6	22%	2175
Difference	0	0	-	-	0	−122,077	−46,408	-	-	0
25% MCED uptake	-	-	-	-	-	61,038	23,204 (4.23)	1:1.6	11%	2175
Difference	0	0	-	-	0	−183,115	−69,613	-	-	0
30% decline in standard-of-care screening, only receiving MCED test	-	-	-	-	-	76,346	30,945 (4.72)	1:1.5	14%	1989
Difference	-	-	-	-	-	−167,807	−61,872	-	-	−186

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
