# Peer review of "Cancer Screening: Present Recommendations, the Development of Multi-Cancer Early Development Tests, and the Prospect of Universal Cancer Screening"

_cancers, 2024, doi:10.3390/cancers16061191_

Round 1

Reviewer 1 Report

Comments and Suggestions for Authors

This is a complex and evolving field so it is helpful that the authors have given an overview of what is going on at this time.

Author Response

We want to express our sincere gratitude for taking the time to review our article. Your insightful comments, particularly regarding the complexity and ongoing evolution of the field, mean a great deal to us. We are thrilled to hear that the overview provided resonated with you and proved to be helpful.

However, in light of all the recommendations from this round of Review, we decided some additions to the manuscript had to be made. As such, we politely invite you to read this improved version. 

Reviewer 2 Report

Comments and Suggestions for Authors

The authors use both MECD and MCED for Multi-Cancer Early Detection.

The possibility of MCED should be explained further convincingly.  More than half of the whole MS deals only with a summary of current tumor biomarkers. It seems that the title 'Universal Cancer Screening – How Close are We?' has not much to do with what addressed in the MS. 

A figure explaining the point of this MS is necessary.

Author Response

  1. We sincerely apologize for this error, and we are grateful for underlining it. We have corrected our manuscript using only the MCED abbreviation for Multi-Cancer Early Detection.
  2. We changed the title to: 

    Cancer screening: Present recommendations, the development of Multi-Cancer Early Development tests, and the prospect of Universal Cancer Screening

  3. In light of the title change, we decided to provide an updated analysis of current screening strategies. Also, we added more information about MCED tests in Table 5. 
  4. More information has been given about manpower, cost-effectiveness (Table 6), and the importance of Artificial Intelligence.
  5. We added Figure 1 in an attempt to illustrate the change from classical screening towards MCED testing. 

Reviewer 3 Report

Comments and Suggestions for Authors

This is a good up to date narrative review on Screening for early detection of cancer. 

1. The title is misleading. There is limited research cited here on Universal cancer screening. It is better to title  this paper as "Cancer screening: Past, present and future" and have an extended subsection on "Universal cancer screening" for the Future 

2. For the cancers discussed, it will help if current national guidelines are given at least from two countries 1. US 2 from Europe. UK has published national guidelines for most screenable cancers.

3. Under other cancers (section 2.6) , more information can be given for sites such as ORAL which has a precancer stage. Many adjunctive tests are discussed in the literature.

4. Cost effectiveness of screening should be presented in more detail. This is what the policy makers are interested in. A supplementary table should include this information and referred to in the text. 

5. The manpower required for screening is not considered. Would Universal Cancer screening be done by family physicians? in community clinics? by health workers?

6. Finally what about AI? Algorithms for suspecting cancers are published. Needs a discussion.  

Author Response

1. We agree, the title wasn't the best. However, we consider the manuscript to be about the present, and the future of cancer screening. Thus, we overhauled our article, and decided the title should be: 

Cancer screening: Present recommendations, the development of Multi-Cancer Early Development tests, and the prospect of Universal Cancer Screening

We are still open to suggestions, however.

2. Thank you very much this suggestion! We decided to overhaul the article and provide updated screening recommendations. We started from Ebell et. al's (cited in the text) idea of a comparison of screening recommendations, and decided to update it from 2018 to 2024. We are convinced that this has been a major improvement towards the Status Quo chapter of our study (see Tables 2, 3, 4). 

3. Your suggestion of us including details about Oral Cancer screening (which is a reasonable request), can run the risk of the article going beyond and above the scope of the article.  Oral cancer does not have any recommendations for screening, and the evidence on efficiency is scarce.  As such, we wanted to see if the direction we were heading with regards to Oral Cancer Screening satisfies your, and if not, we can improve, according to further comments in the second round of review. In our opinion, the most profound lesson from this is that cancer screening doesn't need to be high-tech. Sometimes the best, individualized, patient-tailored screening method is human observation and interpretation. We can discuss more in the article, if the reviewer decides that the direction we chose with this so far suits the article. 

"2.6. Oral Cancer Screening

Visual Oral Examination (VOE) is the recommended test for oral cancer and oral potentially malignant disorders (OPMDs), which implies systematic visual inspection and palpation of the oral cavity under a bright light source, to detect suspect lesions of the oral mucosa [111,112]. Although most national organizations, including USPSTF, do not recommend population-based screening through VOE, an RCT showed a reduction in mortality of 81%, and a reduction of 38% in oral cancer incidence [113] and further studies have demonstrated high sensitivity-specificity for this type of screening [111,114]. What is particular about this form of screening is the non-invasiveness, the low cost associated with screening, and the importance of primary care doctors and dentists, which make the test more accessible, and also lower the burden of manpower needed for population-based screening."

4. Thank you for this remark! We contacted Allan Hackshaw, and he offered us permission to use one of his tables, showing a cost reduction (modelled by machine learning) on how the use of MCED testing could reduce the cost of screening programmes. We believe that this would be very interesting for the policy makers. Also, we made Table 5, offering information about current clinical trials ongoing for MCED tests. We offered information about the manufacturer, the test, the method behind the test. We wanted to include the price, too. However, this was not found for these tests (with the exception of Galleri). Sadly, we can't offer better information than Allan Hackshaw did, this is why we decided to include his table in our manuscript.  

5. Very thoughtful remark, and we want to emphasize that it is relevant! In our literature study, few studies addressed this issue. As such, we added this paragraph in our article:

"5.3. Manpower considerations

It is difficult to estimate if MCED screening would represent a burden or a relief for medical healthcare workers. One of the pivotal requirements of a successful, efficient screening program is a central, well-funded authority, responsible for setting recommendations, and organizing the framework of how these recommendations can be applied (see Tables 2, 3, 4). It is most encouraging to see cancer screening programs offering eligible population tests that do not require healthcare workers (such as FIT) at home [35]. These tests are generally accompanied by greater compliance from the general public because of their non-invasiveness, privacy, and simple nature (see Table 1). Collecting samples such as stool, urine, or saliva, can be sampled by the patient and sent through mail to a specialized laboratory. Though most of the MCED screening tests in clinical trials (see Table 5) require blood samples, this should not be an impediment, as these tests can be offered simultaneously with other regular health check-ups that require this minimally invasive medical procedure. In conclusion, we believe that MCED would not burden already-pressured health services."

Thank you once more for this idea! 

6. Yes, indeed. We added the following paragraph:

"5.5. Bioinformatics and Artificial Intelligence (AI)

             Biomarker research has evolved in close relation with bioinformatic development. Since the development of The Cancer Genome Atlas (TCGA) project, large databases containing valuable data on the multisystem biology features of multiple tumor types led to the development of data mining methods, to propose valuable biomarkers [18,203–209]. Examples include the study of molecular aberrations of the SST gene for multiple GI cancers [210]; the difference in HSP90α protein across 9 different tumors [211]; or the identification of CpG markers to detect 27 cancer types [212].
      Furthermore, the existence of large databases allows machine-learning algorithms to be developed and trained to improve the detection capacity of current biomarkers. Several neural network-based algorithms have been developed to improve detection sensitivity and specificity, as well as tumor of origin (TOO) discrimination [18]. The undeniable benefit of artificial intelligence becomes evident when we observe that the vast majority of MCED tests with ongoing trials have applied deep learning algorithms to improve their diagnostic yield (Table 5). As such, it is important to recognize AI as a key element of biomarker discovery that boosts the output and outcomes of MCED tests."

If the reviewer feels necessary, we can expand on the topic. Thank you! 

Round 2

Reviewer 2 Report

Comments and Suggestions for Authors

Good revision. Accept in its present form.

Reviewer 3 Report

Comments and Suggestions for Authors

Thank you for extensive revisions and incorporating new material.

Comments on the Quality of English Language

subtitles could be more concise